# The experience of intergenerational interactions and their influence on the mental health of older people living in residential care

**Elizabeth Jane Earl** [1]‡ *, **Debbie Marais** [2]‡

**1** Faculty of Medicine and Health Sciences, Stellenbosch University, Cape Town, South Africa, **2** Research Development and Support Division, Faculty of Medicine and Health Sciences, Stellenbosch University, Cape Town, South Africa

‡ These authors contributed equally to this work.

* ejearl1@gmail.com

**Data Availability Statement:** Data cannot be publicly shared because there are ethical restrictions on sharing the complete data set. The data comprises interview transcripts of qualitative,

## Abstract

The mental health of an increasing ageing population is an important part of healthcare. Research has explored means to enrich the lives of older adults living in residential settings, including approaches like the Eden Alternative. This is a cross-sectional, qualitative study with a quantitative component. It looks at common mental health conditions (CMHCs) in residential-living older adults in South Africa and describes their experiences of intergenerational interactions with playschool children. Participants completed a questionnaire which included the Geriatric Depression Scale and Geriatric Anxiety Scale and a semi-structured interview. Anxiety and depression were common in the sample with limited awareness of non-pharmacological therapy available at the facility. The intergenerational interactions were experienced positively with emerging themes of belonging, sense of purpose, reminiscence and positive affective experiences, but influenced by participants' preconceptions of children. The study concludes that intergenerational interactions may serve as adjunctive therapy in managing CMHCs in residential-living older adults. Recommendations are made for successful implementation of such programs.

## Introduction

Globally, the population of older persons is growing [1]. In 2022, 9.2% of South Africa's population was over the age of 60, with this percentage rising continuously [2]. Old age can be defined chronologically or biologically, according to physiological and functional status [3]. In this paper, old age is defined as older than 60, based on South African statistics and factors like life expectancy. In South Africa, life expectancy is 63 (60 in males and 65 in females) [2]. Life expectancy and health are influenced by multiple factors including socio-economic status, gender, and living environments [3]. The proportion of older adults in comparison to the rest of the population in South Africa is lower than in middle- and high-income countries [1]. This, together with the low life expectancy, is due to multiple factors, most notably the HIV/AIDS epidemic [4].

verbatim responses from participants. This is a small group of participants that would be potentially identifiable from their narratives and the way they have spoken about their experiences and involvement in the inter-generational interactions, despite the use of pseudonyms to de-identify the data. Sharing the full data set would violate the principle of confidentiality. However, the thematic analysis with quotes may be provided. For researchers who meet the criteria for access to confidential data, the Health Research Ethics Committee of Stellenbosch University (South Africa) can be contacted to request access to the data (contact: Mrs Shana Tel: 021 938 9657 E-mail: melodys@sun.ac.za).

**Funding:** The authors received no specific funding for this work.

**Competing interests:** The authors have declared that no competing interests exist.

Older persons face the natural ageing process and its associated complications, such as reduced mobility and chronic medical conditions, like arthritis and hypertension [5]. These impact on functional capacity in daily living activities; they are also correlated with a poorer outlook on life, which may lead to common mental health conditions (CMHCs) like anxiety and depression [5].

Ageing is considered a transitional phase of life associated with various adjustments which can be positive or negative [6]. As with health outcomes, the experience of ageing is mediated through physical, psychological, and sociodemographic factors, such as socio-economic status, education, access to healthcare, and sociocultural beliefs and practices associated with being older [7, 8]. This transitional period involves a number of changes that older persons must navigate, including retirement, and dependence [9], which may be exacerbated by a declining sense of agency and autonomy.

Another transition for older adults may entail moving into a residential facility. The reasons for this move may determine the type of facility the older person requires. Retirement homes are living environments for older persons that cater for the changes of ageing. In South Africa, only a minority of older adults reside in such facilities, due to several contextual and economic factors discussed below [4]. Types of retirement home facilities available in South Africa include combinations of independent living (with minimal professional assistance), semi-independent living, and frail care units, providing increasing levels of assistance and nursing care [10]. This study included a semi-independent unit and frail care unit within one retirement facility. In these higher-care facilities, staff may provide care for many older residents and thus standardised routines are developed for residents' daily activities. This may limit the independence of residents.

Older adults, facing challenges like changing roles, reduced physical capabilities, and some loss of independence in retirement homes, may be vulnerable to CMHCs, in addition to the age-related health conditions described above. Studies suggest that, although physical health is well cared for in retirement homes, social and mental health receive less attention [9]. It seems there is a high prevalence of undiagnosed, untreated CMHCs in retirement homes [11].

Multimorbidity and the associated use of multiple medicines (polypharmacy) is highly prevalent, especially in retirement homes [11]; this is medically dangerous and cost-inefficient. Therefore, there is an important role for non-pharmaceutical interventions to improve management of CMHCs [12].

Although many psychiatric conditions can be managed pharmacologically, a combination of medication with non-pharmacological interventions tends to be the recommended standard of care for many mental health conditions, including depression and anxiety [13]. This can include psychotherapy/counselling and various group and physical activities [13]. However, it is also important to identify and understand risk factors for depression and anxiety to prevent/ameliorate these conditions. One intervention is the Eden Alternative (EA), a model for residential care facilities consisting of a set of 10 principles, intended to provide a higher quality of life for residents. It was developed in 1991 by Dr William Thomas [9]. The 10 principles include:

1. Loneliness, helplessness, and boredom are the plaques of the human spirit that account for the bulk of suffering among our elders.

2. Close and continuing contact with plants, animals and children builds a human habitat.

3. Loving companionship is the antidote to loneliness.

4. Giving and receiving care are the antidotes to helplessness.

5. Variety and spontaneity are the antidotes to boredom.

6. Meaning is essential to human life.

7. Medical treatment is a partner in care, not its master.

8. Wisdom grows with honouring and respecting elders.

9. Growth is not separate from life.

10. Wise leadership is the lifeblood of thriving.

The EA is a "philosophy of aged care" [9 p 782] that focuses on minimising factors which exacerbate anxiety and depression in older persons, including "feelings of loneliness, helplessness, and boredom" [14 pp21]. This study focuses on the key principle that describes implementing interactions with 'living things', including children, pets or plants. In doing so, especially through engaging with children, the aim is the development of meaningful relationships and connection to a community [9]. This study explores how intergenerational interactions help fulfil many of the EA principles including developing companionship and producing reciprocal care. These interactions are believed to re-introduce meaning and responsibility to the life of an older person, which may protect against CMHCs [14].

Retirement facilities may improve social engagement between older adults; however, residents may experience a narrowing of their social network, particularly between different age groups [15]. Loneliness experienced by older persons is a risk factor for depression, while social engagement is a protective factor [16]. Loneliness may be ameliorated through intergenerational interactions, as per the EA's approach, thus enhancing social engagement and potentially providing protective mental health benefits. Intergenerational care involves care recipients of different ages, such as residents in frail care units and preschool, in meaningful interaction. It is based on the notion that social relationships are integral to human society, positively influencing mental well-being. There may, however, also be negative outcomes of intergenerational care. Children can be perceived as noisy, energetic, and bothersome [17]. It is thus worth exploring the potential positive and negative effects of intergenerational care on an older population living in retirement facilities.

The residential circumstances of older adults in South Africa vary across different population groups. Most older adults live in multigenerational households within their communities [18]. Reasons for this include the HIV/AIDS epidemic, poverty, and migration [19, 20]; as a result, many older adults live in 'skip generation' households, being the primary caregiver of their grandchildren [4]. In these contexts, intergenerational relationships represent both opportunities and costs for older adults [21].

However, the extended family structure of many South Africans is also influenced by traditional values of *ubuntu*, which defines an individual in relational terms (a person can only be a person through others) [22]. There are direct links between *ubuntu* and intergenerational actions. Sobantu [23 pp99] asserts that all stakeholders need "to address the challenges of older persons such as boredom, loneliness, and illnesses that are common in old age, through . . . the African spirit of *ubuntu* and social development approaches [which] inspire intergenerational human relationships". This suggests that intergenerational programmes, as a non-pharmacological intervention with protective mental health effects, may be well-received within communities. In addition, the Older Persons Act of South Africa (no. 13 of 2006) specifically advocates for the participation of older adults in intergenerational programmes [18, 20].

Some of the drivers for moving into retirement homes include financial considerations, health concerns and reduced mobility, safety concerns, and lack of social and familial support, including emigration of children and grandchildren [24]. The potentially negative effects of

emigration on those left behind are particularly acute when it comes to intergenerational relationship losses [25].

In South Africa, although only a minority move into retirement homes, these older persons may experience the loss of the supportive role provided by their communities. It is valuable, therefore, to explore what intergenerational interactions may add within institutionalised living facilities. Previous research on the outcomes of intergenerational programmes for older adults focused on programmes implemented in community settings [26]. There is limited research in South Africa focusing specifically on the influence of intergenerational care in older individuals living in retirement facilities. This study aimed to develop a further understanding of older adults' experiences living in semi-independent and frail care residential units and how interactions with children may have influenced their mental well-being. As such a programme had already been adopted at the study site, there was an opportunity to explore how residents had engaged with and experienced these interactions, to assess the potential of such programmes as a non-pharmacological intervention with possible mental health benefits.

## Context of study

The study was conducted at the semi-independent and frail care unit of a retirement facility within the Cape Peninsula Organisation for the Aged (CPOA) complex–a South African based non- governmental organisation that develops retirement complexes for older persons. The CPOA manages its retirement homes according to income categories. The facility chosen for sampling is an economic (i.e., for-profit) home but also offers accommodation to middle- to low-income clients. The chosen facility encompasses the EA element of intergenerational care A children's playschool was established on the property in January 2018. Regular, non-compulsory interactive sessions between older persons and children (aged 3–5 years) are held at the frail care unit, and semi-independent residents can join these activities as well. The routinised schedule of the frail care unit facilitates the regular interactions. The interactions happen twice a week, where the children join the older persons for a 60-minute session of playing interactive games such as passing a ball to each other or building puzzles. Furthermore, residents can choose to do additional volunteer work (e.g., reading sessions) with the children. The children also do additional events with the home on special occasions, such as singing songs with them on public holidays. At the time of the study, all participants had been at the facility for more than one year and had been engaging in regular weekly interactions with the children.

## Methods

### Objectives

This study explored the experiences of older persons living in a retirement home who participate in a routine intergenerational programme. The study aimed to assess the benefits and drawbacks of such a programme based on the participants' experiences and possible influence that these experiences may have on their mental well-being. This was guided by the following objectives:

1. To explore the extent of formally diagnosed CMHCs in the sample.

2. To determine the proportion of residents in this sample screening positive for anxiety or depression who are/are not currently on any treatment for a CMHC.

3. To investigate whether sample participants were aware of any non-pharmacological treatments made available for CMHCs at this facility.

4. To explore the residents' experiences of intergenerational interactions with the children.

5. To explore the benefits and drawbacks of these interactions and possible influence on mental well-being.

## Design

This is a cross-sectional, descriptive qualitative study with a quantitative component. For the quantitative component, participants completed a questionnaire assessing features relating to demographics and mental health, as well as two validated screening questionnaires: the Geriatric Anxiety Scale (GAS) [27] and the Geriatric Depression Scale (GDS) [28], which assessed risk of anxiety and depression respectively. The questionnaires provided a baseline assessment of mental health at the facility and the psychological interventions available to residents.

This was followed by a semi-structured qualitative interview. The interview posed questions regarding interactions with the playschool children and how participants felt influenced by them. Each open-ended question had prompts for further elaboration. This structure allowed for an informal conversational style in which the participants could explore their experiences of intergenerational interactions and express any influence on their mental well-being.

## Sampling

A process of non-probability, purposive total population sampling was used, whereby the sample site was determined due to its relevance for the purpose of the study. All residents who met the inclusion criteria were approached. This technique was appropriate as the total possible number of candidates for participation was limited [29]. The participation inclusion criteria were that individuals had to be over the age of 60, living in the chosen retirement home, and to have had some interaction with the playschool children at the facility. Residents who were diagnosed by relevant medical staff as having a neurocognitive disorder were excluded. The total number of residents eligible for participation was 16, of whom five declined to participate and one completed only the questionnaire and not the interview and was thus excluded, leaving 10 participants. The participants of this study are from a range of socio-economic and cultural backgrounds.

## Data collection

After early engagement, once the study obtained ethical clearance, permission for conducting the study at the site was officially given by the CPOA management team, specific facility management and the playschool. Data were collected over a one-month period of June 2019.

Data were collected as follows: eligible residents were individually approached by the primary author, who explained the details of the study and what participation would entail. Individuals were given the opportunity to ask questions and address any concerns. Once consent was received, a time was arranged for participants to complete the questionnaire, either with or without the assistance of the researcher, depending on preference. Most participants chose to do the interview directly after completing the questionnaire. Interviews were conducted by the primary author and were conducted in the participant's private room for ease, comfort and privacy of the participants. The interview consisted of a baseline set of open-ended questions designed to explore the participants experiences, opinions and feelings about engaging with the children. An additional set of questions exploring participants backgrounds and day-to-day interactions were included to assist with good conversational flow. Interviews aimed to last 30 minutes to 1 hour. All the interviews were conducted in English (according to

participant's preference) and all interviews were audio-recorded using a hand-held audio-recording device for transcription.

## Ethical considerations

Ethics approval for this study was granted by the Health Research Ethics Committee at the authors' institution [reference number: U19/04/021]. In accordance with ethical principles, autonomous informed consent was obtained from participants prior to data collection. Confidentiality was protected by using pseudonyms and anonymising feedback to the retirement home. Because older adults in retirement homes, and in frail care units particularly, may experience vulnerabilities, special attention was paid to avoiding coercion and minimising risk to their well-being by participating in the study. The benefits of understanding the experiences of intergenerational care and how this can be used in future were believed to outweigh the minimal risks associated with participation in the study.

## Data analysis

The results of the questionnaires were recorded in three sections: a) demographic data, b) the number of the sample population who self-reported being diagnosed with and/or on treatment for a CMHC; and c) scores from the GAS and GDS. The results were analysed using descriptive statistics. The screening results were recorded according to each scale's scoring method. The GAS categorised results into a *minimal, mild, moderate,* or *severe* risk of anxiety [27]. There is limited data on the sensitivity and specificity of the GAS for a clinical cut-off score, but a score of 16 and above may be used [30]. The scale has a Cronbach alpha of 0.90, indicating good internal consistency [30]. Since the category of mild risk begins at a score of 12 for this screening tool, it was decided that mild, moderate, and severe risk would be evaluated as a potential diagnosis of anxiety. The GDS classified participants as *not suggestive, suggestive,* or *highly indicative* of having depression. For this screening tool, it was decided to take a suggestive or highly indicative score as diagnostic, in the analysis. The Cronbach alpha for the GDS was 0.76, indicating good internal consistency [31]. Each participant's result on the screening scales was compared with the self-reported presence or not of a formal diagnosis of depression and/or anxiety.

The analysis of the interview data followed an iterative, five-step process, as per Braun and Clarke's guidelines on thematic analysis using inductive coding [32]. The initial analysis was conducted by the primary author and development of the framework was checked by both authors for consistency. The first stage was familiarisation with the data, starting with the transcription of the interviews. Ideas emerging from the data were noted and relevant text highlighted. Using an inductive approach, statements that contributed meaningfully towards the objectives of the research, such as mental well-being, intergenerational interactions and so forth, were allocated a descriptive code. The initial codes were then defined and formulated into a coding framework, and the text re-analysed using this framework. The codes were then categorised into overarching themes, which allowed for overlap, omissions, and redundancy across categories to be identified. This analysis was appropriate due to its flexible nature in not following a particular theoretical approach. This allowed for an exploratory approach to developing an inductive analysis of the experiences of intergenerational care and its influence on mental well-being in older persons [32]. This process ensured the transcripts were reviewed and the framework checked for internal consistency across codes and themes, ensuring a coherent analysis [32].

The results of the questionnaire were reviewed in association with the themes that emerged during the analysis of the interviews and examined for possible links between intergenerational care and the mental health of older persons.

**Table 1. Participant demographics.**

|  | Category | Number of participants | Percentage population |
|---|---|---|---|
| **Gender** | Female | 10 | 100% |
|  | Male | 0 | 0% |
| **Age** | 60–69 | 1 | 10% |
|  | 70–79 | 2 | 20% |
|  | 80–89 | 5 | 50% |
|  | 90+ | 2 | 20% |
| **Time living at CPOA home** | 1–2 years | 2 | 20% |
|  | 3–5 years | 5 | 50% |
|  | 6–10 years | 0 | 0% |
|  | >10 years | 3 | 30% |

## Results

### Quantitative

Table 1 displays the demographic information of the ten participants. All were female and half were between the ages of 80 to 89. Fifty percent of the participants had lived at the home for 3–5 years (range: 3 months to 19 years). Table 2 depicts the results of the self-reported CMHC diagnosis, the GAS and GDS results, and the frequency of interacting with the children.

Fig 1 shows that three participants (30%) self-reported being diagnosed with a CMHC; these three were on pharmacological treatment for their mental health condition. Most participants (n = 7) did not report having a diagnosis of anxiety or depression.

Results of the screening scales showed that three participants (30%) were screened to be at mild risk of having anxiety, with none at moderate/high risk. Three participants (30%) screened as suggestive of having depression, and one (10%) was highly suggestive of depression. Four (40%) were screened to possibly have at least one CMHC. All those screened with mild risk of anxiety also scored as suggestive for depression. One participant screened as suggestive of depression with no associated risk of anxiety.

The screening results were compared with participants' self-reported diagnoses of and treatment for depression or anxiety. Two (29%) of those not formally diagnosed and not on treatment (n = 7) screened as potentially at risk for both anxiety and depression (Fig 2). Two

**Table 2. Self-reported CMHC vs screening tool results and frequency of interactions.**

| Participant name | Self-reported diagnosis of CMHC | Geriatric Anxiety Scale | Geriatric Depression Scale | Frequency of engagement with children |
|---|---|---|---|---|
| Holly | Yes, and on treatment | Mild | Suggestive | 3–4 x per week |
| Kate | Yes, and on treatment | Minimal | Suggestive | 3–4 x per week |
| Lily | Yes, and on treatment | Minimal | Not suggestive | 1–2 x per week |
| Olive | No | Mild | Highly indicative | Rarely |
| Rachel | No | Mild | Suggestive | Rarely |
| Tammy | No | Minimal | Not suggestive | 3–4 x per week |
| Janet | No | Minimal | Not suggestive | 1–2 x per week |
| Mary | No | Minimal | Not suggestive | 1–2 x per week |
| Mona | No | Minimal | Not suggestive | 1–2 x per week |
| Pam | No | Minimal | Not suggestive | 1–2 x per week |

Pseudonym used

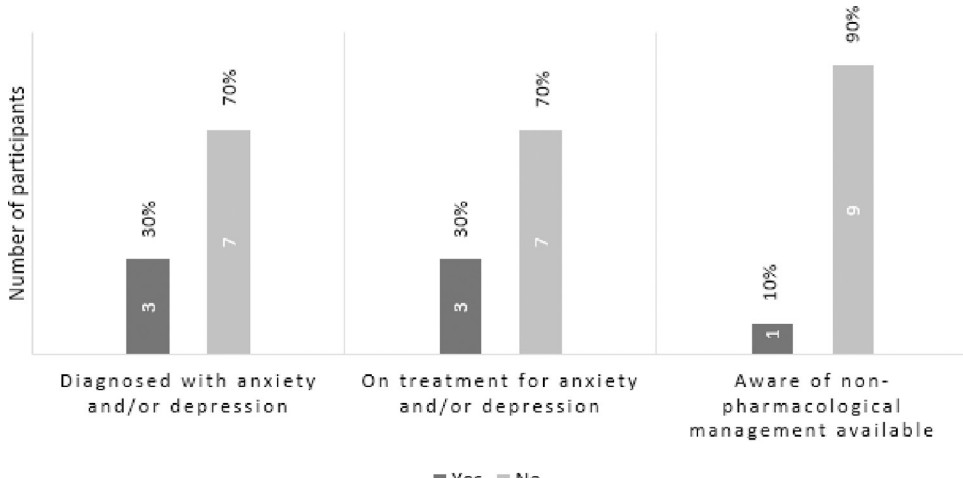

**Fig 1. Self-reported mental health status and treatment.**

of the three participants who reported being diagnosed and on treatment scored positive for either depression or anxiety on the screening scales (Fig 3).

In terms of availability of treatment for mental health conditions, Fig 1 above shows that only one participant was aware of any non-pharmacological interventions available to residents. This participant felt that the activities organised by the facility are therapeutic interventions for psychological well-being, but other residents did not acknowledge these as interventions.

The scores of the screening tools were compared with the frequency of participants' interactions with the playschool children, which is seen in Table 2. Most participants (83%) screening negative for depression or anxiety interacted with the children 1–2 times per week. One participant, who screened negative for both conditions, interacted with the children the most (3–4 times per week). Interestingly, half (n = 2) the participants who screened positive for CMHC

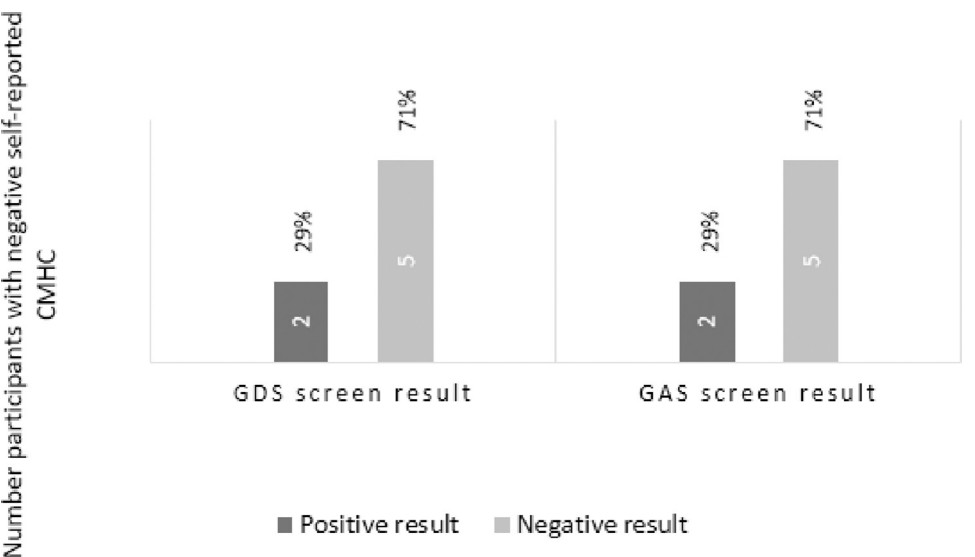

**Fig 2. Association between negative self-reported diagnosis and screening results.**

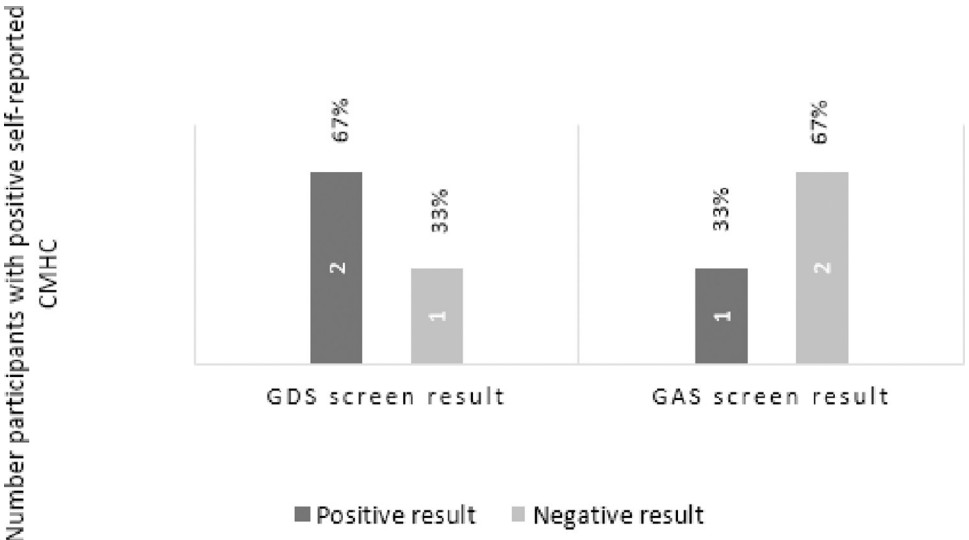

**Fig 3. Association between positive self-reported diagnosis and screening results.**

interacted with the children the most (3–4 times per week), while the other two interacted with the children the least (rarely).

## Qualitative

Themes identified in the interview data are defined in Table 3 and include: preconceptions of children, belonging, a sense of purpose, reminiscence of youth, and affective experience. These themes suggest positive experiences of intergenerational care and the mental well-being of the

**Table 3. Definition of themes and sub-themes.**

| Theme | Definition | Subtheme | Definition |
|---|---|---|---|
| Preconceptions of children | Preconceived ideas about children and the influence this has on the response towards interacting with children | | |
| Belonging | A sense of acceptance and being part of a community of people | Community | Belonging to and having a role to play in a collective of people |
| | | Family | Being part of a smaller functional unit of people |
| | | Individual relationships | Developing connections with individual people |
| Sense of purpose | A feeling of contributing to society | Perceived impact on the children | Having an impact in the lives of others |
| | | Perceived impact by the children | The influence that the children can have on one's life |
| | | Something to look forward to | A desire for a future event that is perceived as pleasant |
| Reminiscence of youth | A fond recollection of the past, specifically relating to one's childhood and the emotional meaning thereof | Playfulness | Youthful engagement and behaving light-heartedly in activities for enjoyment rather than purpose or function |
| | | Nostalgia | A warm longing for and appreciation of the past |
| Affective experience | The influence of an activity on one's mood and how emotional reactions impact self-evaluation and perspective | Upliftment | A sense of enthusiasm and a state of happiness |
| | | Physical affection | A physical relationship with others |
| | | Acceptance | Not being rejected by others |
| | | Endearment | An adoring attitude towards someone |
| | | Irritation | Finding someone to be a nuisance and unpleasant |

participants. These themes, while identified inductively, seem to align with the goals of the EA in minimising loneliness, alleviating boredom, and providing purpose to older persons living in retirement homes. All quotes are given under pseudonyms.

**Preconceptions of children.** This theme explored the participants' preconceived ideas about children. Personal attitudes may affect willingness to engage with children and, considering the non-compulsory nature of the interactions, this could have influenced participants' experiences and responses towards engaging with the children. All participants had had some engagement with the children.

Most participants described a positive attitude towards children throughout their lifetime and the majority reported previous contact with children, mostly with their own grandchildren; however, some reported other means of contact (e.g., as a teacher).

"I have always had a soft spot for a child." (Mona)

In contrast, another participant felt negatively towards engaging with children prior to and during the intergenerational interactions. Her personal experience was as follows:

"I'm just not a small-person person . . . I suppose, yes, a lot of the women especially are very interactive with small children, but I never have been." (Olive)

Additionally, one participant reported that she experienced the interactions more positively than she had expected: "You know, before, you used to think children are just a nuisance, man. But now with these little ones, you learn such a lot. To love them, to see how they hug you and kiss you when they come in here . . ." (Mary)

Regardless of their preconceived attitudes towards children, all residents reported a range of experiences of the intergenerational interactions at the home. The following themes (and subthemes) illuminate how these interactions influenced the residents.

**Belonging.** This theme emerged repeatedly in the interviews. It refers to the sense of acceptance by society and being valued in a community of people. There are three sub-themes within this theme: community, family, and individual relationships.

Participants explored how the interactions with the children created a sense of being part of a community:

"It's something on the side, which makes you feel that you are still part of a varied community, not just lots and lots of old grumpy people . . . The whole of life and creation comes into it. It's just a feeling of global belonging. That you're not an outcast." (Kate)

Participants also felt active in this society.

"I feel a bit involved when I'm with them." (Holly)

The responses also suggested that participants felt a sense of family with these children, indicating a closer bond than just being part of a community. Participants reported benefitting from their perceived roles within this micro-family.

"I like to see them; it makes me feel I'm a grandmother!" (Janet)

"I'm virtually a stranger to them because I'm not part of their family as such . . . And they take to us and they seem to think that we are there in place of their grandparents." (Tammy)

Finally, participants reported valuing the individual relationships that developed through these interactions; these manifested in several ways, including feeling emotional connection:

"You form a sort of relationship . . . You get to know their names and they get to know you. It's a sort of emotional exchange, but a pleasant one." (Kate)

**Sense of purpose.**   This theme addresses the opportunity that the intergenerational interactions provided participants for developing an active role in the children's lives, thereby feeling they can make a valuable contribution to the community. The sub-themes identified included the perceived impact that older persons have had on the children and the perceived impact that the children have had on older persons. It also addresses the sense of having something to look forward to in these interactions.

Participants described various ways in which they believed these interactions have developed the children. For example, the children seemed more sensitive to the needs and limitations of others:

"It shows small children what it can be like for elderly who can't do this and can't do that." (Olive)

In addition to the children's development, these interactions were perceived to impact the well- being of the older residents by giving them a way to contribute to their community. As mentioned, the sense of playing a family-like role in the children's lives is one aspect of this, but participants also felt they developed integral community roles, such as the role as educator:

"It's wonderful to teach a child, especially a young child." (Tammy)

Having these roles and developing purpose in engaging with the children, the older adults were able to learn and develop as well. Mary said she had become "more patient and [able] to consider the children." Kate said:

"It's made me realise how vitally necessary every generation is for every other generation."

As a result of these roles and new opportunities, some participants chose to interact more frequently with the children, in addition to the weekly interactions arranged by the home. This usually took the form of assisting in the children's education at the playschool:

"We were asked if we want to do anything with the children and I put down yes, I want to go read to them. Because a Friday was to me wasted, I wasn't doing anything." (Lily)

This indicates that a change of routine and being relieved of boredom by the children was also important to older persons. This was reiterated by others:

"Takes you out of your own zone here in the frail care." (Holly)

A distinct excitement towards these interactions emerged as a theme. These engagements gave the residents something to look forward to and a sense of purpose, which in turn influenced their well-being at the home:

"But with the children around now, I feel fulfilled, because it's something to look forward to." (Tammy)

**Reminiscence of youth.**    This theme addresses how intergenerational interactions trigger memories and allow for reflection on one's own history, specifically of one's childhood or engagements with children throughout life. It was expressed by participants through a sense of nostalgia and affectionate reflection. Engaging now with children brought about a playful attitude in the residents once again.

> "We play with them like kids, when they come here." (Rachel)

> "That's most important, play, the playing part, you know. It connects you, and the children can communicate, and they come to you freely, you know. It has been real rewarding for me." (Tammy)

As mentioned, these interactions stimulated participants to evaluate their own pasts. This nostalgia was expressed in affectionate tones:

> "They make me miss my own children [chuckles]. They make me look back into the past, which was filled with children [. . .] So, it's nostalgia." (Kate)

**Affective experience.**    This theme explores how the intergenerational interactions influenced the mood and emotional experiences of older persons. It also investigates how the experiences impacted older persons' self-evaluation and perspective through their emotional reactions to the children.

A prominent sub-theme is that of upliftment. Participants reported that seeing the children brought about a shift in mood, highlighting the positive influence the engagements might have on mental well-being:

> "I have noticed that very, very withdrawn, grumpy people actually smile and enjoy [it] when a child comes." (Kate)

> "I think it's just one of the best things for the children living in here, because they're always around. And if you do feel down, you can always go sit with them." (Lily)

The nature of the experiences allowed for a sense of affection to develop between older adult and child, influencing the mood of the engagement. This affection was identified on analysing the descriptions of the uplifting physicality of the interactions.

> "The little ones come in, and then they make a B-line for hugs . . .They come and give you a hug hello and goodbye. Which is really nice. . . I just feel good seeing them." (Holly)

A prominent aspect of ageing is the change in physical appearance that may appear frightening or off-putting to the youth and can lead to older persons feeling dejected, even unloved. The participants explained that, initially, the children were tentative, but this abated with regular interactions. Through the physical nature of the engagements especially, and the concomitant acceptance by the children, self-affirmation was developed in the residents:

> "Because they'll come and run to me and hug me, you know? It makes you feel good, that children don't shy away from you like other children would, you know." (Tammy)

Furthermore, in several interviews, there was a response of affection towards the children. The participants developed a warmth and love for the children through these interactions:

"All of them are so cute." (Lily)

Adding to this sense of endearment was the emotional connection that developed between older adult and child, which created a sense of love and affection in these experiences:

"I think there is a special quality in children that interact with old people who are in their second childhood and it forms a bond–an emotional bond." (Kate)

However, it was identified that there can be negative affective responses in these interactions. One participant, who did not have positive preconceptions about children, felt irritated by the engagements:

"I don't like their screaming coming through my head." (Olive)

This resident chose to join in the sessions with the children only rarely. Even those participants who enjoyed children in general sometimes felt that the children could be a bit boisterous, but felt that this could be ameliorated with good discipline:

"They can make you a bit tired . . . Mind you, at the start, they always seem to come to me to throw the ball. But now I realised I put the ball on my lap and then they stand there and then I push it and then they catch it and now they know." (Mary)

## Discussion

This study explored experiences of intergenerational care, and how it might influence the mental health of older persons living in a frail care and semi-independent unit retirement home in South Africa. South Africa's multi-cultural society means that the experiences of its older generation should not be treated homogeneously. Findings suggest that interactions with children promote a sense of belonging and purpose, evoke reminiscence, and positively influence the mental well-being of older persons. Possible explanations for associations between self-reported diagnoses of CMHCs and screening results are explored. Because data analysis did not include correlational tests for statistical significance, discussion about these associations is merely exploratory and speculative. Thematic findings are discussed in the context of previous research, particularly relating to mental health.

Depression and anxiety have been shown to be prevalent in older persons in general, depression slightly more commonly than anxiety [15]. There is limited research assessing the prevalence of these conditions in residential homes for older persons in South Africa. In this study, the number of symptomatic depression and anxiety in older adults living in residential settings was 40% and 30%, respectively; this is in keeping with international findings [11, 15]. The prevalence of CMHCs across all age groups in South Africa is 9.7% [33]. Despite the small sample group, our study indicates a high number of CMHCs in comparison to the general population.

The prevalence of CMHCs in older adults can be explained by several risk factors that this population may face. Many older persons live with a higher rate of chronic conditions which have been associated with poorer mental health [5]. Institutional living may be an independent risk factor for CMHCs, but this has been linked to an increase in risk factors for depression,

such as loneliness and reduced family connection [15]. Other risk factors for CMHC include the loss of a societal role, autonomy and independence [9].

Prevalence results for CMHCs only partially corresponded with participants' self-reported diagnosis of and treatment for a CMHC in this study. Of the four participants who screened positive for either depression or anxiety, two self-reported a diagnosis of and treatment for these conditions. It may be that the two participants who screened positive but did not self-report had an as yet undiagnosed CMHC. Studies show that CMHCs may be under-diagnosed in older persons living in residential settings [11]. However, given the small sample size and lack of statistical associations, limited conclusions can be drawn.

Another possible explanation for CMHCs in residential settings may be the inadequate availability of interventions. Only one participant in this study noted availability of non-pharmacological management of CMHCs for the residents at the facility. This could be due to lack of education as to what constitutes non-pharmacological therapies, as the facility does provide activities hosted by an occupational therapist. It is therefore important to explore and raise awareness about non-pharmacological interventions that may be effective in ameliorating risk for CMHCs, such as the intergenerational interactions investigated in this study. The influence of these intergenerational interactions on older persons' experiences of their lives and their mental well-being emerged in the qualitative themes discussed.

## Preconceived ideas

This study suggests that a positive attitude towards children generally is associated with a more enthusiastic response towards interactions with children in the retirement home. Research suggests that a positive attitude is often held by older persons towards young people, especially if they have had more contact with children [34]. This seems to be confirmed by the findings of this study. In their review of intergenerational programmes, Galbraith et al. [35] revealed a lack of critical reflection on how older persons' preconceptions of children might influence the results of intergenerational interactions, but note that it may be a significant factor in limiting the transferability of intergenerational interventions [35]. Assessing pre-existing attitudes towards an intervention may be an important precursor to implementing successful interventions.

## Belonging

A sense of belonging (within community, family, and individual relationships) was identified as an important aspect of participants' interactions with the children. The idea of belonging through the development of relationships in older persons is recognised in the literature, particularly through interactions with children [36]. Belonging in an individual relationship can signify a sense of worth to the person involved. In this study we see how the individual relationship between child and older adult was valued by older persons, as they felt important through being remembered individually by the children.

The participants in this study described developing a feeling of familial connection with the children. A loss of family connection can occur when moving into a residential home and can be a risk factor for poorer mental health [15]. Grandparents throughout history have played significant developmental and supportive roles in the lives of children, especially in the South African context [8]. In developing a relationship with children in residential environments, older persons living at these facilities are potentially reciprocally fulfilling the role of grandparent, reconnecting with their sense of family belonging [8].

Engaging with the children also reminded the participants of the greater community to which they belong. The idea of belonging is emphasised in the South African concept of

*ubuntu* [22]. The intergenerational interactions seemed to have the effect of evoking a feeling of global belonging among our study participants, suggesting a positive association with the concept of *ubuntu*.

This sense of belonging may have a protective effect for mental health in older persons living in residential settings. In providing a platform for socialising and developing a sense of belonging, intergenerational interactions have the potential to reduce loneliness (a CMHC risk factor) and have a positive impact on the mental health of older persons [17]. This seems to have been the case for the participants in our study.

## Sense of purpose

Participants in our study expressed a sense of purpose in their interactions with the children, specifically through various roles with which they identified, as discussed above. A loss of role identity can occur in older persons, especially when moving into retirement homes where the options for community involvement may become limited [36]. The intergenerational interactions provided a platform in which our participants felt they could contribute and play a role in the community (e.g., teaching the children) [17]. Momtaz et al. [37] indicate improved perceived physical and psychological well-being of their older participants through gaining the opportunity to provide support or play a role in the lives of others [37].

Participants in this study chose to engage in the programme. Having a meaningful activity in which to participate provided the residents with an opportunity to volunteer involvement in this community. This may be important as an expression of self-agency and autonomy [38], factors that can be limited by residential living, which can contribute to poorer mental health in retirement settings [9]. This concept is not necessarily associated only with interacting with children and could occur through a variety of volunteer-based services and interactions.

The interactions also provided our participants with a space for self-development. The older persons in this study felt that they were learning through the interactions (e.g., developing patience). The idea of both generations developing together has been shown to be significant in making interactions meaningful in their impact on the well-being of older persons [35]. In the South African context, this is part of the *ubuntu* philosophy, which has been explicitly linked to intergenerational relationships [23]. Bohman et al. [6 pp450] posit that "reciprocal care" is significant to the support of older persons and intergenerational relationships.

Having an activity to look forward to in the retirement home also emerged as a benefit of interacting with the children in this study. This may partly be because the interactions provide relief from the day-to-day residential routine. Monotony may be associated with residential living which can lead to boredom and reduction in quality of life [9]. Assessing level of boredom is part of the GDS, so alleviation of boredom may help to alleviate symptomatic depression [28]. Ultimately, this relief of boredom, and the development of enthusiasm, correlates with the goals of the EA and suggests an improvement in the well-being of the residents [14]. Another explanation for the enthusiasm is the light-heartedness and playfulness that the interactions with children facilitate.

## Reminiscence of youth

In this study, interacting with children seemed to stimulate the residents and, through their own behaviour, they began to reflect the youthful attitude of the children. Youthful upliftment has been described as beneficial in intergenerational interactions [17]. Through these interactions, older persons may forget about their ailments and aged appearance and what these imply (e.g., chronic diseases); thereby, intergenerational interactions decrease older persons' identification with a sickly, frail role [39].

The participants reported that acting youthfully stimulated self-reflection about their past; this included reminiscing about their youth and caring for their own children; this may provide affirmation of their past [8]. The nostalgia associated with the interactions seemed positive in this study and is reflected in similar research [17]. This reflection and affirmation can lead to self-acceptance, possibly promoting resilience against the adversities of ageing, including CMHCs [16]. Furthermore, the process of reminiscing may enhance self-esteem and promote a sense of successful ageing [40]. Interactions with children, specifically, seem to facilitate reminiscing, which has been used in therapy as a protective tool against CMHCs in older persons [40].

## Affective experience

The theme of affective experience captured how the participants reacted emotionally towards the interactions. Findings showed that the emotional response to the experience of intergenerational interactions was generally positive, with a positive shift in mood, consistent with previous research [39]. Psychological research has shown that interventions inducing positive feelings can be associated with an improvement in mental health, especially if the intervention is continued regularly [41]. It must be noted, however, that the literature also indicates that people with depression do not always sustain a positive affect in response to positive stimuli [42]. Therefore, engagement in intergenerational interactions by older persons may promote positive affective change, benefitting mental health, provided that it is done regularly.

One reason for this positive response may be the affectionate relationships developed through the interactions. The residents were more expressive around the children and their mood shifted. This may be specific to engaging with children; Xaverius and Mathews [43] noted previously that levels of expression and engagement increase when older persons engage with children.

In addition to *being* affectionate, the participants perceived this to be reciprocated by the children. This manifested through physical affection and how the children did not shy away from them, despite their own perceptions of appearing undesirable in their old age. The nature of the interactions (whether active or passive) did not seem to impact the themes, but the physical affection between child and adult during the interactions was significant to participants. This suggests that these interactions facilitate a sense of being accepted by the children. Being accepted and included despite appearances is a strong element of this experience and potentially aids in boosting self-esteem; this seems to occur in older persons who engage with children [35]. Enhanced self-esteem has been associated with perceived improved mental and physical health [37].

Participants indicated that interacting regularly with the children improved their enjoyment of the interactions, as the children learnt how to behave around the older adults. This facilitated the formation of relationships with the children. Therefore, developing a regular programme and ensuring appropriate preparation for the intergenerational interactions may be important in producing satisfactory effects.

## Limitations

This study had a small number of participants due to the population size involved in the interactions with the playschool, limiting the transferability of these findings. Due to time limitations, a longitudinal study was not possible, and the effect of the intergenerational interactions on the prevalence of symptomatic CMHCs over time cannot be commented on. Furthermore, the self-reported diagnosis of the participants did not differentiate between anxiety and depression, limiting comparisons with screening results. All the participants were female, reflecting the limited number of male older residents living in the home.

## Conclusions and recommendations

This study noted a high number of CMHCs among older persons living at this facility, with limited provision of non-pharmacological interventions in the management thereof. Research looking at the well-being of older adults living at shared site facilities is rare in South Africa, and this study indicates that our findings mirror global findings. Intergenerational interactions may serve as an adjunctive non-pharmacological intervention for CMHCs in older persons living in residential homes. By forming relationships with the children, older persons felt connected to a community and developed a sense of belonging in society. They re-identified with roles which gave them a sense of purpose and they valued their contribution to the lives of the children. In this study, the intergenerational interactions provided participants who have moved out of traditional roles as community grandparents with the ability to embody such roles again. Interestingly, the literature indicates that 'family' members are not necessarily actual relatives in the South African context [6]. It is often accepted that an older person in the community may care for various other community members as family. Looking at the positive response seen in these intergenerational interactions with non-family members shows how the South African context is well-suited to these programmes as there is already an acceptance of, and connectedness with, non-family members.

Furthermore, a youthful attitude and reminiscences seemed to enhance positive self-evaluation and self-acceptance and evoked a sense of playfulness. Intergenerational interactions also appeared to bring about a positive mood shift in older persons. These factors could be protective against CMHC in older adults and have the potential to contribute to the temporary alleviation of symptomatic depression and anxiety by reducing risk factors associated with CMHCs in the older population.

For future research, it is recommended that a larger, longitudinal study (with both male and female participants) be performed to assess the transferability of the beneficial effects of intergenerational interactions on older persons. To assess impact on mental health, standard measures of mental health indicators should be used. Research that explores the impact of intergenerational interactions in retirement homes versus older persons living in communities would illuminate factors within these environments that may enhance the effects of such interactions. Furthermore, future research could explore how various levels and frequencies of engagement could impact experiences of intergenerational programmes. It may also be valuable to explore the influence of intergenerational interactions on children.

Based on the results of this study, recommendations for the implementation of intergenerational interactions can be made. There should be appropriate training of staff involved in the interactions, to ensure preparation of both older persons and the children for the engagements [44]. Due to the influence of preconceived ideas of children, interactions should be voluntary, which facilitates residents' agency. It is recommended that interactions be run according to a regular programme to promote the formation of relationships and bonding between participants [39]. The regular activity also provides older persons with an activity to look forward to, alleviating boredom. Activities organised for the interactions should allow for reciprocal engagement and facilitate an educational or skills-based development to promote a sense of purpose in older persons through the interactions.

## Author Contributions

**Conceptualization:** Elizabeth Jane Earl, Debbie Marais.

**Data curation:** Elizabeth Jane Earl.

**Formal analysis:** Elizabeth Jane Earl.

**Investigation:** Elizabeth Jane Earl.

**Methodology:** Elizabeth Jane Earl.

**Project administration:** Elizabeth Jane Earl, Debbie Marais.

**Resources:** Elizabeth Jane Earl, Debbie Marais.

**Software:** Elizabeth Jane Earl.

**Supervision:** Debbie Marais.

**Validation:** Elizabeth Jane Earl, Debbie Marais.

**Visualization:** Elizabeth Jane Earl.

**Writing – original draft:** Elizabeth Jane Earl.

**Writing – review & editing:** Debbie Marais.

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
