## [Decision Letter · Decision Letter 0]

6 Dec 2022

PONE-D-22-20338The experience of intergenerational interactions and their influence on the mental health of older people living in residential carePLOS ONE

Dear Dr. Elizabeth,

Thank you for submitting your manuscript to PLOS ONE. After careful consideration, we feel that it has merit but does not fully meet PLOS ONE’s publication criteria as it currently stands. Therefore, we invite you to submit a revised version of the manuscript that addresses the points raised during the review process.

The reviewers' comments are at the end of this e-mail. 

We look forward to receiving your revised manuscript.

Kind regards,

Pracheth Raghuveer, MD, DNB

Academic Editor

PLOS ONE

Journal Requirements:

2. Please include a separate caption for each figure in your manuscript

Reviewers' comments:

Reviewer's Responses to Questions

**Comments to the Author**

1. Is the manuscript technically sound, and do the data support the conclusions?

Reviewer #1: Yes

Reviewer #2: Yes

2. Has the statistical analysis been performed appropriately and rigorously? 

Reviewer #1: Yes

Reviewer #2: I Don't Know

3. Have the authors made all data underlying the findings in their manuscript fully available?

Reviewer #1: Yes

Reviewer #2: No

4. Is the manuscript presented in an intelligible fashion and written in standard English?

Reviewer #1: Yes

Reviewer #2: Yes

5. Review Comments to the Author

Reviewer #1: thank you for the opportunity to review the manuscript. It is important to see how Eden Alternative is being implemented in South Africa considering the growing ageing population over there and globally. I have two comments for the authors to consider:

1. The Eden Alternative has got quite a few principles but from what I read the key aspect this article/study has focused on is the intergenerational care to engage with older people to be more connected. I do wonder whether the authors need to provide a bit more information on the background of the EA and be specific on the aspects this study has focused on. Given the intergenerational care is the key theme, it may be worthwhile to expand this a bit more in the literature review.

2. I can see that one of the objectives was to determine the prevalence of formally diagnosed CMHCs in the sample. Initially I thought the sample of the study was quite big but then only realised there were only 10 participants. I would be cautious to say the objective was to determine the prevalence but more on 'explore the extent of formally diagnosed CMHCs in the sample".

Reviewer #2: Commendable work. Small sample size and absence of male participants are the two glaring limitations, as already acknowledged by you. Number of weeks for which the interactions between the residents and the children took place has not been clearly mentioned. A longitudinal assessment would definitely be appreciable. Age range of the children was not mentioned clearly, though I understand that they belong to the playschool group. Another thing could have been mentioned that which type of interaction, whether active physical or passive ones were more closely associated with the themes mentioned.

6. PLOS authors have the option to publish the peer review history of their article (what does this mean?). If published, this will include your full peer review and any attached files.

Reviewer #1: No

Reviewer #2: No

---

## [Author Response · Author response to Decision Letter 0]

15 Dec 2022

Journal requirements

The manuscript has been formatted accordingly. 

2. Please include a separate caption for each figure in your manuscript 

A caption has been included for each figure at the point where the figure should be inserted in the manuscript, according to the format requirements provided. 

The reference list has been reviewed and is complete and correct. 

 

Reviewer 1: 

1. The Eden Alternative has got quite a few principles but from what I read the key aspect this article/study has focused on is the intergenerational care to engage with older people to be more connected. I do wonder whether the authors need to provide a bit more information on the background of the EA and be specific on the aspects this study has focused on. Given the intergenerational care is the key theme, it may be worthwhile to expand this a bit more in the literature review. 

We have expanded further on the background of the Eden Alternative in the literature review, including listing the principles. We highlight here that intergenerational interactions are part of the EA in their promotion of interacting with “living things” which is what this study explores. We note that exploring intergenerational interactions we can see how they fulfill principles of the EA by promoting community, companionship and meaning, thus improving the quality of life of older residential living adults. In this study, we explore how this in turn may impact the mental well-being of older adults. (Introduction: p. 3-4)

2. I can see that one of the objectives was to determine the prevalence of formally diagnosed CMHCs in the sample. Initially I thought the sample of the study was quite big but then only realised there were only 10 participants. I would be cautious to say the objective was to determine the prevalence but more on 'explore the extent of formally diagnosed CMHCs in the sample". 

The objective has been reviewed and adjusted as recommended. (Methods: objectives P. 7)

Reviewer 2: 

1. Small sample size and absence of male participants are the two glaring limitations, as already acknowledged by you. 

We agree that these limitations are significant. Research that is able to address these limitations will improve our knowledge gaps in this field. 

2. Number of weeks for which the interactions between the residents and the children took place has not been clearly mentioned. A longitudinal assessment would definitely be appreciable. 

The participants had all resided there for more than 1 year and interactions had been occurring during this time period. The study was conducted over 1 month. In table 1, the length of time participants have lived at the facility is included, and the the context of the study we mention that the playschool has been established there since 2018 (a total of one and a half years). The data collection occurred over a one month period during which participants were interacting regularly with the children. As mentioned in our limitations, the time constraint in conducting the study prevented longitudinal research, but is definitely worth exploring in the future. (Context of study: p. 6 and data collection: p. 8)

3. Age range of the children was not mentioned clearly, though I understand that they belong to the playschool group. 

The age range of the children is 3-5 years old. As the children were not the focus group of this study, we do not have data on their demographics. However, this age range is as per the playschool’s stipulations. The playschool does accept younger age groups as well (age 6 months-3 years) who do join for some of the interactions when there is enough supervision, but not necessarily all of them. We have included the regular age range (3-5) in the context of the study. (Context of study: p.6)

4. Another thing could have been mentioned that which type of interaction, whether active physical or passive ones were more closely associated with the themes mentioned.

The participants engaged in both active and passive activities (mentioned in the context of the study) but this did not affect our themes. What did impact the themes was that regardless of the nature of interaction, the children were usually physically affectionate towards the older adults. A key finding was how the adults enjoyed the affection provided by the children as it meant they were not rejected or found to be undesirable by the children. (Discussion: p.26)

---

## [Decision Letter · Decision Letter 1]

23 Mar 2023

PONE-D-22-20338R1The experience of intergenerational interactions and their influence on the mental health of older people living in residential carePLOS ONE

Dear Dr. Earl,

Thank you for submitting your manuscript to PLOS ONE. After careful consideration, we feel that it has merit but does not fully meet PLOS ONE’s publication criteria as it currently stands. Therefore, we invite you to submit a revised version of the manuscript that addresses the points raised during the review process.

We look forward to receiving your revised manuscript.

Kind regards,

Pracheth Raghuveer, MD, DNB

Academic Editor

PLOS ONE

Journal Requirements:

Reviewers' comments:

Reviewer's Responses to Questions

**Comments to the Author**

1. If the authors have adequately addressed your comments raised in a previous round of review and you feel that this manuscript is now acceptable for publication, you may indicate that here to bypass the “Comments to the Author” section, enter your conflict of interest statement in the “Confidential to Editor” section, and submit your "Accept" recommendation.

Reviewer #1: All comments have been addressed

Reviewer #3: All comments have been addressed

2. Is the manuscript technically sound, and do the data support the conclusions?

Reviewer #1: Yes

Reviewer #3: Yes

3. Has the statistical analysis been performed appropriately and rigorously? 

Reviewer #1: N/A

Reviewer #3: Yes

4. Have the authors made all data underlying the findings in their manuscript fully available?

Reviewer #1: Yes

Reviewer #3: No

5. Is the manuscript presented in an intelligible fashion and written in standard English?

Reviewer #1: Yes

Reviewer #3: Yes

6. Review Comments to the Author

Reviewer #1: Thank you for the revised manuscript. The authors have addressed the comments adequately. I recommend the manuscript to be considered for publication.

Reviewer #3: Comments to authors

This paper reports very important findings that can be useful inform policy and practice to help older population. The authors have carefully addressed comments from the reviewers and the manuscript has improved significantly. I have some further minor comments to be addressed by the authors.

Introduction

Lines 17-18: “In 2011, 8% of South Africa’s population 18 was over the age of 60, with this percentage rising continuously [2].”

This is very old data, reported more than years ago. I suggest the authors provide updated or current data on this.

Data collection

Provided details about interview venue, who did the interviews, interview questions, duration of interviews, recordings or fieldnotes, etc….

Data analysis

Please clearly describe the steps undertaken during qualitative data analysis and how the authors ensured the reliability and quality of the data.

7. PLOS authors have the option to publish the peer review history of their article (what does this mean?). If published, this will include your full peer review and any attached files.

Reviewer #1: No

Reviewer #3: No

---

## [Author Response · Author response to Decision Letter 1]

5 Apr 2023

Response:

The reference list has been reviewed.

The second reference: “2. South Africa. Statistics South Africa. Vulnerable groups series II: The social profile of older persons SA, 2011 - 2015 [Internet]. Pretoria: The Department; 2017. Available from http://www.statssa.gov.za/publications/Report%2003-19-03/Report%2003-19-032015.pdf” [2]

has been removed and replaced with a relevant, current reference: “2. Statistics South Africa. Marginalised Groups series Volume VI: The social profile of older persons SA, 2017 - 2021 [Internet]. Pretoria: Statistics South Africa; 2023. Available from https://www.statssa.gov.za/publications/03-19-08/03-19-082021.pdf” [2]

Reviewer 1: 

Thank you for the revised manuscript. The authors have addressed the comments adequately. I recommend the manuscript to be considered for publication.

Response:

Thank you for this assessment. 

Reviewer 3:

1. Introduction

Lines 17-18: “In 2011, 8% of South Africa’s population 18 was over the age of 60, with this percentage rising continuously [2].”

This is very old data, reported more than years ago. I suggest the authors provide updated or current data on this.

Response: 

Thank you for highlighting this outdated statistic. At the time of initial writing the latest statistics had not yet been released. It has been updated accordingly in the manuscript and the citation adjusted with the most recent published statistics. (Introduction p.1)

2. Data collection

Provided details about interview venue, who did the interviews, interview questions, duration of interviews, recordings or fieldnotes, etc….

Response:

The details mentioned have been added to the text as noted. The primary author conducted the interviews in the participants’ private rooms. The interview questions were open-ended questions designed to explore the participants experiences, opinions and feelings about engaging with the children. An additional set of questions were available to ease conversational flow and discuss participants backgrounds and day-to-day interactions. Interviews were recorded using a hand-held audio-recorder for transcription at a later stage. (Data collection p.9)

3. Data analysis

Please clearly describe the steps undertaken during qualitative data analysis and how the authors ensured the reliability and quality of the data

Response:

We have elaborated on the steps undertaken during the analysis of the qualitative data as advised. It is divided into multiple steps of which all were cross-checked by both authors for reliability and quality of the data. In summary the steps taken included the following:

1. Familiarization with the interview data

2. Identifying emerging ideas that relate or contribute to the objectives.

3. Ideas are allocated descriptive codes which are then defined. 

4. Coding framework is developed and the text re-analysed 

5. Codes are categorised into overarching themes,

6. Re-analysed text for any overlaps/omissions etc

7. All steps reviewed by both authors 

(Data analysis p.10)

---

## [Decision Letter · Decision Letter 2]

5 Jun 2023

The experience of intergenerational interactions and their influence on the mental health of older people living in residential care

PONE-D-22-20338R2

Dear Dr. Earl,

We’re pleased to inform you that your manuscript has been judged scientifically suitable for publication and will be formally accepted for publication once it meets all outstanding technical requirements.

Kind regards,

Pracheth Raghuveer, MD, DNB

Academic Editor

PLOS ONE

Additional Editor Comments (optional):

Reviewers' comments:

Reviewer's Responses to Questions

**Comments to the Author**

1. If the authors have adequately addressed your comments raised in a previous round of review and you feel that this manuscript is now acceptable for publication, you may indicate that here to bypass the “Comments to the Author” section, enter your conflict of interest statement in the “Confidential to Editor” section, and submit your "Accept" recommendation.

Reviewer #1: (No Response)

2. Is the manuscript technically sound, and do the data support the conclusions?

Reviewer #1: Yes

3. Has the statistical analysis been performed appropriately and rigorously? 

Reviewer #1: N/A

4. Have the authors made all data underlying the findings in their manuscript fully available?

Reviewer #1: Yes

5. Is the manuscript presented in an intelligible fashion and written in standard English?

Reviewer #1: Yes

6. Review Comments to the Author

Reviewer #1: Looking at the other reviewers' comments, I think the authors have now addressed all the essential components.

7. PLOS authors have the option to publish the peer review history of their article (what does this mean?). If published, this will include your full peer review and any attached files.

Reviewer #1: No

---

## [Editor Report · Acceptance letter]

8 Jun 2023

PONE-D-22-20338R2 

The experience of intergenerational interactions and their influence on the mental health of older people living in residential care 

Dear Dr. Earl:

I'm pleased to inform you that your manuscript has been deemed suitable for publication in PLOS ONE. Congratulations! Your manuscript is now with our production department. 

Kind regards, 

on behalf of

Dr. Pracheth Raghuveer 

Academic Editor

PLOS ONE